# Input-Aware Dynamic Backdoor Attack

**Tuan Anh Nguyen**[1,2]**, Tuan Anh Tran**[1,3]
[1]VinAI Research, [2]Hanoi University of Science and Technology, [3]VinUniversity
`v.anhnt479@vinai.io`, `v.anhtt152@vinai.io`

## Abstract

In recent years, neural backdoor attack has been considered to be a potential security threat to deep learning systems. Such systems, while achieving the state-of-the-art performance on clean data, perform abnormally on inputs with predefined triggers. Current backdoor techniques, however, rely on uniform trigger patterns, which are easily detected and mitigated by current defense methods. In this work, we propose a novel backdoor attack technique in which the triggers vary from input to input. To achieve this goal, we implement an input-aware trigger generator driven by diversity loss. A novel cross-trigger test is applied to enforce trigger nonreusablity, making backdoor verification impossible. Experiments show that our method is efficient in various attack scenarios as well as multiple datasets. We further demonstrate that our backdoor can bypass the state of the art defense methods. An analysis with a famous neural network inspector again proves the stealthiness of the proposed attack. Our code is publicly available.

## 1 Introduction

Due to their superior performance, deep neural networks have become essential in modern artificial intelligence systems. The state-of-the-art networks, however, require massive training data, expensive computing hardwares, and days or even weeks of training. Therefore, instead of training these networks from scratch, many companies use pre-trained models provided by third-parties. This has caused an emerging security threat of neural backdoor attacks, in which the provided networks look genuine but intentionally misbehave on a specific condition of the inputs.

BadNets [1] is one of the first studies discussing this problem on the image classification task. The authors proposed to poison a part of the training data. More specifically, the poisoned images were injected with a fixed small pattern, called a trigger, at a specific location, and the corresponding labels were changed to some pre-defined attack classes. The trained networks could classify the clean testing images accurately but quickly switched to return attack labels when trigger patterns appeared. Liu et al. [2] extended it to different domains, including face recognition, speech recognition, and sentence attitude recognition. Since then, many variations of backdoor attacks have been proposed [3, 4].

Though varying in mechanisms and scenarios, all these attacks rely on the same premise of using a fixed trigger on all poisoned data. It became a crucial weakness that has been exploited by defense methods [5, 6, 7, 8]. These methods derive backdoor trigger candidates then verify by applying them to a set of clean test images.

We argue that the fixed trigger premise is hindering the capability of backdoor attack methods. A dynamic backdoor that has the trigger pattern varying from input to input is much stealthier. It breaks the foundation assumption of all current defense methods, thus easily defeats them. Moreover,

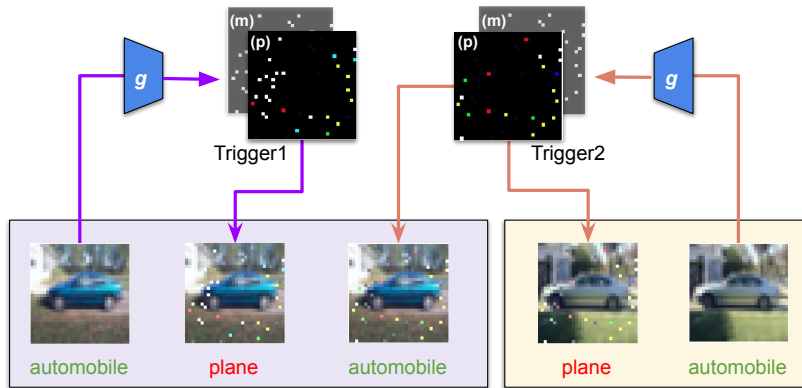

Figure 1: **Input-aware Dynamic Backdoor Attack.** The attacker uses a generator $g$ to create a trigger $(m, p)$ conditioned on the input image. The poisoned classifier can correctly recognize clean inputs (the leftmost and rightmost images) but return the predefined label ("plane") when injecting the corresponding triggers (the second and fourth image). Triggers are nonreusable; inserting a trigger to a mismatched clean input does not activate the attack (the middle image).

dynamic triggers are hard to differentiate from adversarial noise perturbations, which are common in all deep neural networks.

For simplification, we limit our work on image recognition problems. However, it could be easily extended to many other domains, such as speech recognition and sentence attitude recognition, as proposed in [2].

Implementing such input-aware backdoor systems is not trivial. Without careful design, the generated triggers may be repetitive or universal for different input images, making the systems collapse to regular backdoor attacks. To construct such a dynamic backdoor, we propose to use a trigger generator conditioned on the input image. The generated triggers are distinctive so that two different input images do not share the same trigger, compelled by a diversity loss. Also, the triggers should be non-reusable; a trigger generated on one image cannot be applied to another image. To enforce this constraint, we define a novel cross-trigger test alongside the traditional clean and poisoned data tests and incorporate the corresponding loss in training. Fig. 1 illustrates the proposed attack system.

We evaluate our method by running experiments on the standard datasets MNIST, CIFAR-10, and GTSRB. On every dataset, the backdoor model shows near a 100% attack success rate on poisoned data while it accurately classifies the clean and cross-trigger images to the benign labels. The trained models also easily break the state-of-the-art defense methods, including Neural Cleanse, STRIP, and Fine-Pruning, proving the effectiveness of the proposed attack.

We further investigate potential practices against our attack. First, a simple image regularization such as spatial smoothing or color depth shrinking may impair the trigger pattern and break down the attack. However, we show that our backdoor is persistent to these regularizations. Second, we test the standard network inspection tool GradCam [9] over our trained models. There is no visible trail, again proving the stealthiness of the proposed backdoor.

## 2 Background

### 2.1 Threat model

Backdooring is a technique of injecting a hidden malicious functionality into a system. This functionality is activated only when a trigger appears. The attacker can secretly exploit the backdoor to gain illegal benefits from the system. For instance, a backdoored face authentication system may allow any person with a specific accessory to access all user accounts.

We consider the standard threat model used by most backdoor attack and defense studies. The adversary has total control over the training process, and purposely trains the deep neural network with a backdoor. The infected network is then provided to the customer as is. The customer can run protection methods before or after deploying the model.

## 2.2 Previous Backdoor Attacks

We first review BadNets [1], the most common backdoor attack method. The network is trained for an image classification task $f : \mathcal{X} \to \mathcal{C}$, in which $\mathcal{X}$ is an input image domain and $\mathcal{C} = \{c_1, c_2, ..., c_M\}$ is a set of $M$ target classes. A clean training set $\mathcal{S} = \{(x_i, y_i) | i = \overline{1, N}\}$ is provided, in which $x_i \in \mathcal{X}$ is a training image and $y_i \in \mathcal{C}$ is the corresponding label.

A backdoor trigger $t = (m, p)$ consists of a blending mask $m$ and a pattern $p$. To ensure stealthiness, the trigger mask must be small, i.e. $\bar{m} \ll 1$. During training, the training sample $(x, y)$ is randomly replaced by a poisoned one $(\hat{x}, c)$ with a probability $\rho$. The poisoned image $\hat{x}$ is constructed from the clean input $x$ and the trigger $t = (m, p)$ through an injecting function $\mathcal{B}$:

$$\hat{x} = \mathcal{B}(x, t) = x \odot (1 - m) + p \odot m \tag{1}$$

where $\odot$ is pixel-wise multiplication. The attack label $c \in \mathcal{C}$ can be fixed (single-target attack) or a function of the ground-truth label $y$ (all-to-all attack). A successfully trained backdoor model will accurately classify clean images while returning the predefined attack label $c$ when the trigger $t$ appears. These scenarios are called clean test and attack test, respectively.

After BadNets, many different attack variations have been proposed. Chen et al. [3] suggested to blend accessories, e.g., glasses, as backdoor triggers, making physical attacks possible. Liu et al. [2] offered an effective backdoor injection at fine-tuning instead of the training stage. To reduce fine-tuning effort, a reverse engineer technique was introduced to search for an efficient pattern $p$ from the pre-trained model $f$. This work also extended the attack to various domains such as speech recognition. Ji et al. [10] generated backdoor patterns conditioned on the target class. Lately, Salem et al. [11] extended the trigger to a set of patterns at a set of locations making it more dynamic and stealthy. Still, all proposed methods rely on uniform backdoor triggers regardless of **input images**.

## 2.3 Backdoor Defenses

Since backdoor attacks are an emerging concern, backdoor defense has become crucial. Many studies on this topic have been proposed in recent years. We can classify them into three categories regarding usage: training defense, model defense, and testing-time defense.

### 2.3.1 Training defense

This defense assumes the defender has access to training data. Therefore, it focuses on data anomaly detection [12]. This assumption, however, does not match our scenario. The backdoor model is intentionally trained by a malicious third-party. We, therefore, will skip this defense approach.

### 2.3.2 Model defense

Model defense methods attempt to verify or mitigate the third-party model before deployment. Fine-Pruning [13] tried to neutralize the deep network by cutting neurons that are dormant on clean inputs. It, however, could not verify if a model had backdoors. Neural Cleanse [5] was the first work that could detect poisoned model. For each label $l$, Neural Cleanse computed the minimal trigger candidate $t$ that could convert any clean image to $l$. It then detect among these candidates the abnormally smaller ones as backdoor indicators. ABS [6], instead, scanned the neurons and generated trigger candidates via reverse engineering technique. These triggers were then verified by being applied to a set of clean images. Cheng et al. [14] used GradCam [9] to analyze the network behavior on a clean input image with and without the synthesized trigger to detect the anomalies. Recently, Zhao et al. [15] indirectly applied mode connectivity into examining backdoor behaviors, effectively mitigating backdoor while maintaining acceptable models' performance on benign data.

Except for Fine-Pruning and Mode Connectivity approach, all the defense methods above assumed the backdoor trigger was image-independent. The trigger candidates in Neural Cleanse were optimized on all clean inputs, while the synthesized triggers in other techniques needed verifying on different reference images.

### 2.3.3 Testing-time defense

We can also defend at testing-time when the deep model is already deployed. The task now is to verify if the provided image is poisoned and how to mitigate it.

STRIP [7] exploited the persistent outcome of the backdoor image under perturbations for detection. More specifically, it superimposed various image patterns on the input and expected the predictions would be random for benign model/input but consistent under attack scenarios. In contrast, Neo [8] searched for the candidate trigger patches where region blocking led to prediction change. Recently, Doan et al. [16] used GradCam inspection to detect potential backdoor locations. In all these methods, the trigger candidates were then verified by being injected into a set of clean images. This practice, again, strongly relied on the uniform trigger assumption.

## 3 Method

We argue that a universal backdoor trigger for all images is a bad practice and an Achilles heel of the current attack methods. The defender can estimate that global trigger by optimizing and verifying on a set of clean inputs. Hence, we propose a new method, in which each image has a unique trigger, and that trigger of an image will not work on other images.

### 3.1 Definition

Recall that a classifier is a function $f : \mathcal{X} \rightarrow \mathcal{C}$, in which $\mathcal{X}$ is a input image domain and $\mathcal{C} = \{c_1, c_2, \ldots, c_M\}$ is a set of $M$ target classes. In order to train the function $f$, a training dataset $\mathcal{S} = \{(x_i, y_i) | x_i \in \mathcal{X}, y_i \in \mathcal{C}, i = \overline{1, N}\}$.

Poisoning the classifier $f$ requires poisoning the training dataset. Let $\mathcal{B}$ be the injecting function, applying triggers $t = (m, p)$ to clean images:

$$\mathcal{B} : \mathcal{X} \times \mathcal{T} \longrightarrow \mathcal{X}$$
$$(x, t) \longmapsto \mathcal{B}(x, t) = x \odot (1 - m) + p \odot m$$

where $m$ is a mask and $p$ is a pattern.

Instead of using a fixed uniform trigger $t$, we propose a strictly input-aware backdoor attack.

**Definition 1.** *A backdoor attack is **input-aware** if and only if any trigger $t$ is a function of the corresponding clean input $x$. Denote $g$ as the pattern generator that derives $t$, mapping from input image domain to pattern domain:*

$$g : \mathcal{X} \longrightarrow \mathcal{P}$$
$$x \longmapsto g(x) = t$$

**Definition 2.** *An input-aware backdoor attack is **strict** if and only if the trigger generated by $g$ satisfies:*

- ***Diversity:*** *$\exists \epsilon > 0$ such that*

$$\frac{\|g(x_j) - g(x_i)\|}{\|x_j - x_i\|} > \epsilon \qquad \forall i \neq j,$$

- ***Nonreusablity:*** *A trigger generated for one input image cannot be used on another*

$$f(\mathcal{B}(x_i, g(x_j))) = y_i \qquad \forall i \neq j.$$

Note that based on the first criterion, $g$ is Reverse Lipschitz continous. Maximizing $\frac{\|g(x_1) - g(x_2)\|}{\|x_1 - x_2\|}$ will prevent $g$ from being saturated, mapping $\mathcal{X}$ into a significant subset of $\mathcal{P}$. Therefore, $g$ could be able to generate patterns notably different from input to input.

In the next sections, we will discuss how to implement the new attack method. First, we design the trigger generator as an auto-encoder working on clean image input. Second, we propose a novel cross-trigger test, alongside the standard clean and attack tests, to enforce trigger nonreusability. Finally, we sum up the objective functions used in training.

### 3.2 Trigger generator network

The trigger generating network $g$ has a basic encoder-decoder architecture. It takes an image as an input, and generates a pattern correspondingly. After that, the generated pattern will combine with a pre-defined mask and the original input image (to create a backdoor input) or another image (used for enforcing the nonreusability of trigger).

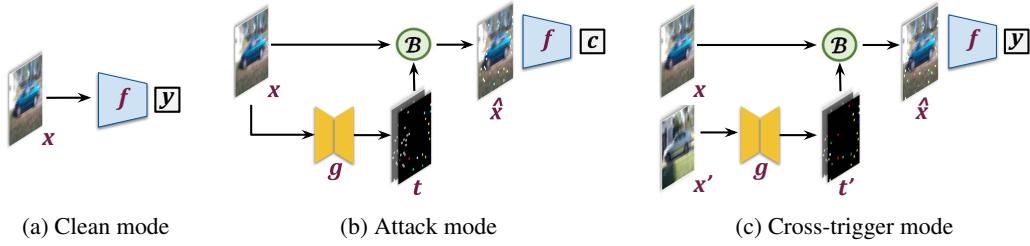

<center>(a) Clean mode        (b) Attack mode        (c) Cross-trigger mode</center>

<center>Figure 2: Three running modes of the proposed input-aware backdoor system.</center>

## 3.3 Running modes

Existing backdoor attacks are trained on two modes: (a) **clean mode** in which the network has to correctly recognize clean images and (b) **attack mode** in which the attack is activated on poisoned data. To enforce trigger nonreusablity, we propose in training a novel **cross-trigger mode**. Given a clean input $x$, we randomly select a different image $x'$ and generate the corresponding trigger $t'$. The injected image $\mathcal{B}(x, t')$ is assigned to the clean label $y$. These modes are illustrated in Fig. 2.

In evaluation, we have three tests accordingly: clean test, attack test, and cross-trigger test.

## 3.4 Objective functions

**Classification loss**. Our classifier loss will be computed from 3 running modes. We first select the backdoor probability $\rho_b \in (0, 1)$ and the cross-trigger probability $\rho_c \in (0, 1)$ such that $\rho_b + \rho_c < 1$. Then, for each clean input $(x, y)$, we randomly select the mode and compute loss accordingly. Then, we sum up to get the classification loss:

$$\mathcal{L}_{cla} = \sum_{(x,y)\in\mathcal{S}, x'\in\mathcal{X}, x'\neq x} \begin{cases} \mathcal{L}(f(\mathcal{B}(x, G(x))), c) & \text{with probability } \rho_a \\ \mathcal{L}(f(\mathcal{B}(x, G(x'))), y) & \text{with probability } \rho_c \\ \mathcal{L}(f(x), y) & \text{with probability } 1 - \rho_a - \rho_c \end{cases}, \quad (2)$$

where $\mathcal{L}$ is the cross-entropy loss function.

**Diversity loss**. To update the pattern generator $g$, along with classification loss $\mathcal{L}_{cla}$, we also use the *diversity enforcement regularisation* $\mathcal{L}_{div}$:

$$\mathcal{L}_{div} = \frac{\|x - x'\|}{\|g(x) - g(x')\|}$$

This regularisation is the prerequisite for training the generator $g$. Without this regularisation, the output of $g$ will saturate to a uniform trigger.

The final objective function will be the sum of $\mathcal{L}_{cla}$ and $\mathcal{L}_{div}$, with a weighting parameter $\lambda_{div}$:

$$\mathcal{L}_{total} = \mathcal{L}_{cla} + \lambda_{div}\mathcal{L}_{div}$$

Algorithm 1 illustrates the training pipeline of our input-aware backdoor attack.

# 4 Experiments

## 4.1 Experimental Setup

Following the previous backdoor papers, we conducted experiments on the MNIST [18], CIFAR-10 [19] and GTSRB [20] datasets. Detailed information of these datasets are reported in Table 1. To build the classifier on CIFAR-10 and GTSRB, we used Pre-activation Resnet-18 [17], as suggested by [21]. The MNIST classifier has a self-defined structure as presented in Table 1. As for the generators, we use simple network architectures that are detailed in Fig. 3a.

We use the SGD optimizer for training classifier $f$, and Adam optimizer for training generator $g$ with the same learning rate 0.01. This rate drops 10 times after every 100 epochs. The networks are jointly trained until converged. We use $\lambda_{div} = 1$ and $\rho_b = \rho_c = 0.1$ in our experiments.

<center>5</center>

**Algorithm 1:** Strictly input-aware backdoor attack training

---

1  Let $f$ be the classifier, $g$ be the trigger generator;
2  Given a target label $c$, a training dataset $\mathcal{S}$, backdoor probability $\rho_b$, cross-trigger probability $\rho_c$;
3  **initiate** $f, g$
4  **for** *number of training iterations* **do**
5      **for** $(x, y)$ *in* $\mathcal{S}$ **do**
6          $d \longleftarrow random(1); (x', y') \longleftarrow random\_sample(\mathcal{S})$
7          $t \longleftarrow g(x); t' \longleftarrow g(x')$
8          $\mathcal{L}_{div} \longleftarrow {\|x - x'\|}/{\|g(x) - g(x')\|}$
9          **if** $d < \rho_b$ **then**
10             $\mathcal{L}_{cla} \longleftarrow \mathcal{L}(\mathcal{B}(x, t), c)$
11         **else if** $d < \rho_b + \rho_c$ **then**
12             $\mathcal{L}_{cla} \longleftarrow \mathcal{L}(\mathcal{B}(x, t'), y)$
13         **else**
14             $\mathcal{L}_{cla} \longleftarrow \mathcal{L}(x, y)$
15         **end**
16         $\mathcal{L}_{total} = \mathcal{L}_{cla} + \lambda_{div}\mathcal{L}_{div}$
17         $g \longleftarrow optimize_g(\mathcal{L}_{total}); f \longleftarrow optimize_f(\mathcal{L}_{total})$
18     **end**
19 **end**
20 **return** the trained models $f, g$

---

Table 1: Detailed information of the datasets and the classifiers used in our experiments. Each convolution (conv) and fully-connected (fc) layer is followed by a ReLU, except the last fc layer.

| Dataset | Subjects | #Labels | Input Size | #Train. Images | Classifier |
|---------|----------|---------|------------|----------------|------------|
| MNIST | Written digits | 10 | $28 \times 28 \times 1$ | 60000 | 2 conv, 2 fc |
| CIFAR-10 | General objects | 10 | $32 \times 32 \times 3$ | 50000 | PreActRes18 [17] |
| GTSRB | Traffic signs | 43 | $32 \times 32 \times 3$ | 39252 | PreActRes18 [17] |

### 4.2 Attack experiments

We conduct our experiments in the common single-target attack, in which the target label $c$ is the same for all images. Some sample backdoor images are presented in Fig. 4, and the results on testing sets are reported in Fig. 3b. For all three datasets, the backdoor attack success rates (ASR) are almost 100%, while still achieving the same performance on clean data as the benign models do. Moreover, the cross-trigger accuracy are from 88.16% (CIFAR-10) to 96.80% (GTSRB), proving the backdoor trigger inapplicable on unpaired clean images. Note that our backdoor classifier, just by studying on training data, can recognize unseen backdoor patterns generated from unseen test images. This generalization ability is very important feature of our method. We also reported the multi-label attack experiments in the supplementary.

### 4.3 Defense experiments

We are now testing our attack approaches against the current backdoor defenses, in both **Model defense** and **Testing defense** scenarios.

#### 4.3.1 Model Defenses

We first verify our backdoor algorithm using Neural Cleanse [5], Fine-Pruning [13], and Mode Connectivity [15]. They are representatives for three different model-defense approaches: pattern optimization, neuron analysis, and model repairing based on loss landscapes.

Neural Cleanse computes the optimal patterns to convert all clean inputs to each target label. It then checks if any pattern is significantly smaller than the others as the backdoor indicator. Neural Cleanse quantifies it by the Anomaly Index metric and uses $\tau = 2$ as the clean/backdoor threshold. As can be seen in Fig. 5a, our system easily passed this test. We can explain by the fact that the backdoor trigger varies from image to image, so there is no such small pattern that can activate the attack on all inputs.

| Layers | # Channels | |
| --- | --- | --- |
| | MNIST | Others |
| (ConvBlock) x 2, maxpool | 16 | 32 |
| (ConvBlock) x 2, maxpool | 32 | 64 |
| (ConvBlock) x 2, maxpool | - | 128 |
| ConvBlock, upsample, ConvBlock | - | 128 |
| ConvBlock, upsample, ConvBlock | 32 | 64 |
| ConvBlock, upsample, ConvBlock | 16 | 32 |
| ConvBlock, sigmoid | 1 | 3 |

| Dataset | Clean | Attack | Cross |
| --- | --- | --- | --- |
| MNIST | 99.54 | 99.54 | 95.25 |
| CIFAR-10 | 94.65 | 99.32 | 88.16 |
| GTSRB | 99.27 | 99.84 | 96.80 |

(a) Generator networks        (b) Attack experiments

Figure 3: Network architecture of the generators and performance of the trained models. Each ConvBlock consists of a Conv2D (kernel 3x3), a BatchNorm, and a ReLU layer. The final ConvBlock does not have ReLU.

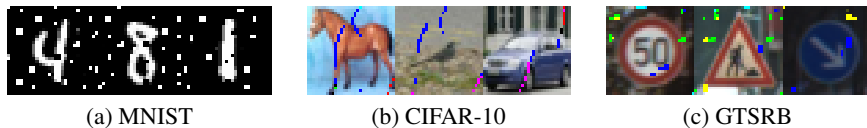

(a) MNIST        (b) CIFAR-10        (c) GTSRB

Figure 4: Sample backdoor images generated from our networks. The target label for each set is digit-0 (MNIST), airplane (CIFAR-10), and speed-limit-20 (GTSRB).

In contrast, Fine-Pruning focuses on neuron analyses. It gradually removes the dormant neurons that are conditioned on clean images from a specific layer to mitigate the model. We test Fine-Pruning on our backdoor models. The plots in Fig. 5a presents the model accuracy in attack and clean mode with respect to the number of neurons pruned. On all datasets, at no point is the clean accuracy substantially higher than the attack one, making backdoor mitigation impossible.

Lately, Mode Connectivity suggests to mitigate backdoor by using a curve model lying on a path connection between two tampered models in the loss landscapes. The model connection is optimized based on a small set of clean training samples. The curve model is located on that path connection by a parameter $t \in [0, 1]$; in which $t = 0$ corresponds to the first tampered model and $t = 1$ corresponds to the second one. This work claims that the curve model formed with an appropriate $t$, e.g., 0.1, will have an acceptable clean accuracy while no longer be affected by the backdoor. We run that method on our backdoor models on CIFAR-10 and report the error rates in Fig. 6. As can be seen, the backdoor error rate is always close to the clean one. The largest gap between them is only 20% when using 2500 training samples and t = 0.5. Hence, this defense method cannot mitigate our backdoor.

### 4.3.2 Testing-time Defenses

We also verify our method by STRIP [7], a representative of the testing-time defense methods. Given an input image potentially poisoned, STRIP will perturb it through a random set of clean images and monitor entropies of the prediction outputs. A traditional backdoor attack is persistent to such perturbation, leading to stable and low entropy. The proposed method, however, deactivates the attack mode when image content changes and mismatches the trigger. Hence, it produces high prediction entropies and covers a similar entropy range as that of the clean model, visualized in Fig. 5c.

## 4.4 Ablation studies

### 4.4.1 The necessity of cross-trigger test and diversity loss

To demonstrate the efficacy of our objective functions, we train different versions of the classifier $f$ in the CIFAR-10 dataset without these loss functions.

**Without the cross-trigger test**. While achieving the same performance on the clean and attack test, the cross-trigger accuracy of the trained classifier is only around 10 %. The trigger generated for one image could be applied to any other inputs, causing $f$ evident to standard backdoor detectors. Indeed, we verified $f$ with Neural Cleanse, and it was easily caught with Anomaly Index 9.43.

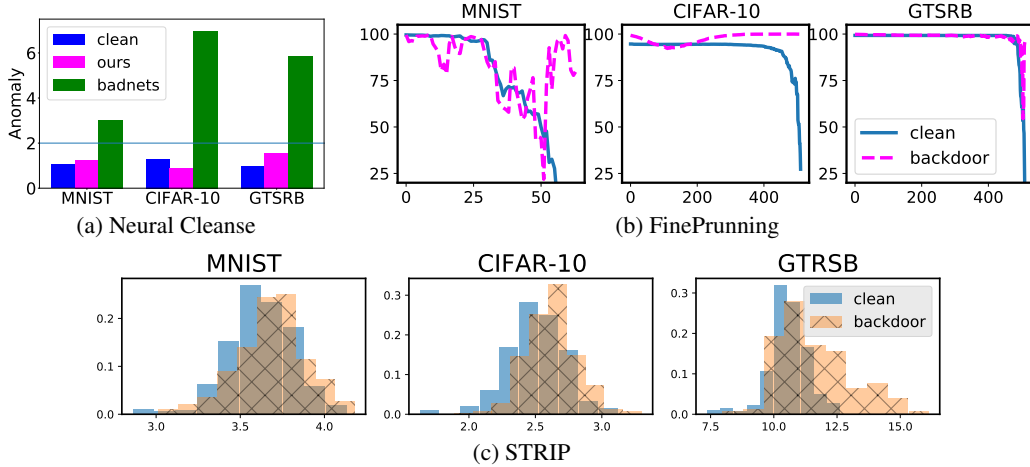

Figure 5: Experiments on verifying our backdoor models by the state-of-the-art defense methods.

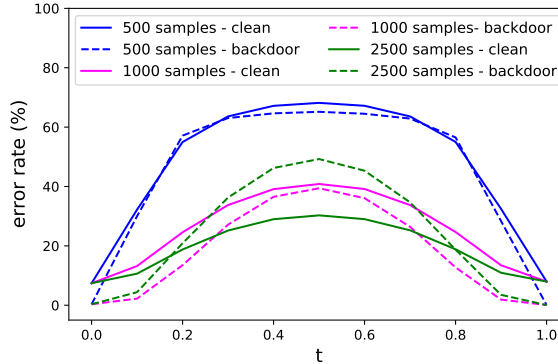

Figure 6: **Mode Connectivity experiments.** This plot shows the curve model's clean and attack error rates with different value $t$ on the CIFAR-10 dataset. The model connections are fine-tuned with small numbers of clean samples.

**Without the diversity loss**. The generator $g$ outputs the same pattern for any inputs. The cross-trigger and attack mode are now contradicting as $f$ is forced to output the true class and target class with the same input. Therefore, the classifier $f$ fails to converge, having all clean, backdoor, and cross accuracy stuck at 10%.

**Without both losses**. The generator $g$ outputs the same pattern for any inputs. Although the model can converge, it behaves like a normal BadNet model.

#### 4.4.2 Analysis on backdoor probability $\rho_b$ and cross-trigger probability $\rho_c$

In this experiment, we will study how the hyper-parameters $\rho_b$ and $\rho_c$ affect the backdoor model's clean, backdoor, and cross accuracy. To do so, we train multiple backdoor models on the CIFAR-10 dataset with either $\rho_c$ or $\rho_b$ varying from 0.025 to 0.5.

As can be seen in Fig. 7, our method is quite stable with high accuracy on clean, attack, and cross tests. When increasing $\rho_c$, the cross accuracy increases from 80% to 93%. When increasing $\rho_a$, the attack success rate goes up to near 100%, and the cross accuracy also surprisingly increases.

### 4.5 Behaviour analyses

#### 4.5.1 Image regularization

According to [22], the perturbation based attacks may be vulnerable to simple image regularization processes such as image smoothing or color depth shrinking. We track the clean and attack accuracy

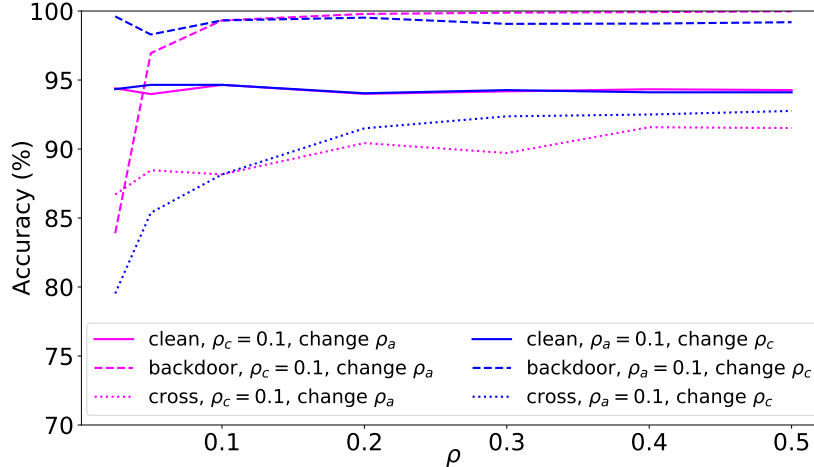

Figure 7: Model clean, attack, and cross accuracy on the CIFAR-10 dataset when changing $\rho_b$ and $\rho_c$.

| Test | Original | Smoothing | | Color shrinking | |
|---|---|---|---|---|---|
| | | k = 3 | k = 5 | 3 bits | 1 bit |
| Clean | 94.65 | 69.30 | 35.24 | 85.81 | 22.87 |
| Attack | 99.32 | 99.61 | 99.67 | 98.76 | 89.94 |

(a) Image regularization

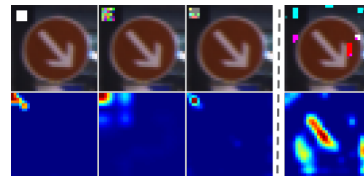

(b) GradCam

Figure 8: **Behaviour analyses on our backdoor models:** (a) Clean and attack accuracy under different image regularizations in CIFAR-10, (b) Comparison between previous backdoor attacks [1, 10, 11] (first three columns) and ours (last column), presented by the poisoned GTSRB input (top) and the output heatmap (bottom).

of the trained CIFAR-10 backdoor model when applying those regularizations and report results in Fig. 8a. Surprisingly, the proposed system is robust to these pre-processes. While the clean accuracy quickly drops, the attack accuracy still stays at 90-100%.

### 4.5.2 Network inspection

Traditional backdoor attacks heavily rely on image-agnostic triggers; thus, they can be exposed under a network inspection such as GradCam [9] as reported by recent studies [14, 16]. We examined backdoor models from Badnets and two recent attack methods [10, 11] via GradCam visualization on their poisoned GTSRB images, and the triggers were easily caught as shown in Fig. 8b. We then verified our model and visualized its result in the final column. The highlight regions spread out instead of focusing on the trigger, avoiding any inspection based defenses. We can explain it by the fact that our network has to match the trigger to the image content, making them equally important.

## 5 Conclusion and future works

In this paper, we have presented a novel backdoor attack that is conditioned on the input image. To implement such a system, we use a trigger generator generating triggers from the clean input images. We enforce the generated triggers to be diverse and nonreusable for different inputs. These strict criteria make the poisoned models stealthy to pass through all defense practices. It raises the backdoor threat to a higher bar for future security research.

While being effective, our system can be further improved. The current trigger patterns are unnatural, so we aim to make them more realistic and imperceptible to humans.

## Broader Impact

Our work is beneficial for both the research community and practical AI systems.

For the research community, our work points out the weakness of the current backdoor research, both attack and defense studies, when heavily relying on the assumption of fixed and universal triggers. It raises the backdoor threat to a higher bar for future security research.

With the practical AI systems, our work will raise awareness of deep models' security. It points out a potential advanced backdoor inside the deep-learning-based components acquired from third-parties. People, therefore, can look for potential protection against backdoor exploitation. It is particularly crucial to the security systems.

Certainly, the attacker can also gain benefit from our work to design such effective backdoor models. Still, we believe novel and efficient defenses against the proposed attack will soon be introduced after our research released.

## Funding Disclosure

This research is funded by VinAI Research.

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
