[Supplementary Material · Input_Aware_Dynamic_Backdoor___Supplentary_material.pdf]

# Input-Aware Dynamic Backdoor Attack
## — Suplementary material —

## Abstract

Besides the main paper, we also provide a PDF and code as supplementary materials. This PDF contains detailed information about the datasets and the systems that we used in the experiments. It then provides more examples of the generated backdoor images and GradCam inspection results in single-target attack experiments. Finally, we present our experiments on all-to-all attack scenarios, in which multiple target labels are used.

## 1 System details

### 1.1 Datasets

We conduct our experiments in 3 datasets, from simple to more complex ones. Importantly, these datasets are all used in numerous previous works in both attack and defense studies. This makes our results more comparable and reliable.

#### 1.1.1 MNIST

The dataset [1] is a subset of a larger set available from the National Institute of Standards and Technology (NIST). It consists of 70,000 grayscale images of handwritten digits at resolution $28 \times 28$. The dataset is divided into a training set of 60,000 images and a test set of 10,000 images. It could be found at `http://yann.lecun.com/exdb/mnist/`.

During the training step, we randomly apply random cropping to the input image. No augmentation is applied in the evaluation stage.

#### 1.1.2 CIFAR-10

CIFAR-10 [2] is a labeled subset of the 80-millions-tiny-images dataset, collected by Alex Krizhevsky, Vinod Nair, and Geoffrey Hinton. The dataset consists of 60,000 color images at resolution $32 \times 32$ of 10 different classes, with 6,000 images per class. It is splitted into two sets: the training set of 50,000 images and the test set of 10,000 images. The dataset is public and available in `https://www.cs.toronto.edu/~kriz/cifar.html`.

We apply random crop, random rotation, and random horizontal flip during the training process. In the evaluation stage, no augmentation is applied.

#### 1.1.3 GTSRB

The German Traffic Sign Recognition Benchmark - GTSRB [3] dataset is originally from a challenge held at the International Joint Conference on Neural Networks (IJCNN) 2011. This dataset contains

30 more than 50,000 images with 43 classes. Image sizes vary from $15 \times 15$ to $250 \times 250$ pixels. It com-
31 prises a training set of 39,209 images and a test set of 12,630 images. The GTSRB dataset is publicly
32 available at `http://benchmark.ini.rub.de/?section=gtsrb&subsection=dataset`.

33 In both training step and evaluation step, images are resized into $32 \times 32$ pixels. Input images are
34 then applied random crop and random rotation in the training procedure. No augmentation is applied
35 during the evaluation procedure.

## 1.2 Classifiers

### 1.2.1 MNIST

38 We use a simple convolution net for this dataset. Detailed architecture will be mentioned below.

Table 1: Detailed architecture of MNIST classifier. * means the layer is followed by a Dropout.

| Layer | Filter | Filter Size | Stride | Activation |
|---|---|---|---|---|
| Conv2d* | 32 | $5 \times 5$ | 1 | ReLU |
| Maxpool | 32 | $2 \times 2$ | 2 | - |
| Conv2d* | 64 | $5 \times 5$ | 1 | ReLU |
| Maxpool | 64 | $2 \times 2$ | 2 | - |
| Linear* | 1024 | - | - | ReLU |
| Linear | 10 | - | - | Softmax |

### 1.2.2 CIFAR-10 and GTSRB

40 For the CIFAR-10 and GTSRB datasets, we use PreActRes18 [4] structure for the classifier.

## 1.3 Trigger mask and pattern generators

42 For simplicity, in the paper, we mentioned the trigger generator $g$ as a whole. In practice, we divided
43 it into two networks $g_m$ and $g_p$ for trigger mask generation and trigger pattern generation. These
44 networks have the same structure as presented in the paper except the number of output channels; $g_m$
45 returns a single-channel output mask.

46 The pattern generator $g_p$ is jointly trained with the classifier, as described in the paper. To make the
47 training process stable, we pre-train the mask generator $g_m$, then freeze it during the main training
48 process.

## 1.4 Training details

50 We use Adam optimizer for training mask generator $g_m$, with the learning rate of $0.01$. This learning
51 rate drops 10 times each 10 epochs. We train $g_m$ for 25 epochs, then freeze this module for continue
52 training classifier $f$ and pattern generator $g_p$.

53 We use Adam optimizer for training pattern generator $g_p$ with the learning rate of $0.01$. Learning rate
54 of $g_p$ drops 10 times for each 100 epochs since the $200^{th}$ epoch. For the classifier $f$, we use SGD
55 optimizer with the learning rate of $0.01$. The learning rate of $f$ reduces by 10 times every 100 epochs.
56 We train $g_p$ and $f$ simultaneously and until the models converge at $\sim 600$ epochs.

## 1.5 Running time

58 We use a system of a GPU RTX 2080Ti and a CPU i7 9700K to conduct our experiment. Detailed
59 inference time of each module will be demonstrated below.

Table 2: Inference time of our modules.

|  | Classifier ($f$) | Mask generator ($g_m$) | Pattern generator ($g_p$) |
|---|---|---|---|
| MNIST | $6.51\,ms$ | $0.94\,ms$ | $0.99\,ms$ |
| CIFAR-10 | $7.96\,ms$ | $1.36\,ms$ | $1.41\,ms$ |
| GTSRB | $7.88\,ms$ | $1.36\,ms$ | $1.42\,ms$ |

## 2 Additional experiment results

### 2.1 Sample backdoor images

We present sample backdoor images generated by our systems, in comparison to the ones from the traditional BadNet models, in Fig. 1. As can be seen, our backdoor trigger varies from image to image.

### 2.2 Network inspection

We present more GradCam inspection results of our backdoor networks on GTSRB and CIFAR-10 dataset in Fig. 2.

## 3 All-to-all attack

Besides the single-target attack scenario, we also verify the effectiveness of input-aware dynamic backdoor attack in multi-target scenario, often called all-to-all attack. In this scenario, the input of class $y$ would be targeted into class $y + 1$.

### 3.1 Experimental Setup

We use the same experimental setups as in the single-target scenario, with a small modification. In the attack mode at training, we replace the fixed target label $c$ by $y + 1$. In the attack test at evaluation, we also change the expected label similarly.

### 3.2 Sample backdoor images

We present sample backdoor images generated by our systems, in comparison to the ones from the traditional BadNet models, in Fig. 3.

### 3.3 Attack experiments

We conducted attack experiments and reported results in Table **??**. While models still achieve state-of-the-art performance on clean inputs, the attack efficacies slightly decreases. This is due to the fact that the target label now varies from input to input. Though, the lowest attack accuracy is 93.16%, which is still pretty high.

Similar to the single-target scenario, the triggers are still nonreusable, proved by the cross-trigger accuracies.

Table 3: **All-to-all attack result.**

| Dataset | Clean | Attack | Cross |
|---|---|---|---|
| MNIST | 99.46 | 98.47 | 94.34 |
| CIFAR-10 | 94.49 | 93.16 | 89.40 |
| GTSRB | 98.93 | 98.13 | 94.29 |

(a) MNIST

(b) CIFAR-10

(c) GTSRB

Figure 1: **Sample backdoor images**. The first row are clean images. The second and the third rows are badnet's patterns and badnet's images. The fourth and the final rows are our patterns and backdoor images.

(a) GTSRB

(b) CIFAR-10

Figure 2: **All-to-one attack**: Backdoor inputs and their corresponding heatmaps.

## 3.4 Defense experiments

We repeat the same defense experiments used in the single-target scenario. Our backdoor models can pass all the tests as can be seen in Fig. 4, 5, and 6.

## 3.5 Behaviour analyses

### 3.5.1 Image regularization

Similar to the single-target scenario, we apply different image regularization techniques on the CIFAR-10 backdoor model trained for the all-to-all attack, as shown in Fig. **??**. In all tests, the clean and attack accuracy are pretty similar, negating backdoor mitigation attempts.

Table 4: **Effect of image regularization on the CIFAR-10 backdoor model, all-to-all attack scenario.**

| Test | Original | Spatial smoothing | | Color depth shrinking | | |
|---|---|---|---|---|---|---|
| | | k = 3 | k = 5 | 3 bits | 2 bits | 1 bit |
| Clean | 94.49 | 68.37 | 34.76 | 86.56 | 63.60 | 28.22 |
| Attack | 93.16 | 66.63 | 35.29 | 85.38 | 62.26 | 25.18 |

### 3.5.2 Network inspection

Finally, we present the GradCam inspection results of our all-to-all backdoor networks on GTSRB and CIFAR-10 dataset in Fig. 7. The heatmaps, again, spread over the input image, failing to catch the backdoor regions.

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

(a) MNIST

(b) CIFAR-10

(c) GTSRB

Figure 3: **Sample all-to-all backdoor images**. The first row are clean images. The second and the third rows are triggers and backdoor images, respectively.

Figure 4: Neural Cleanse

Figure 5: Fine-pruning

Figure 6: STRIP

(a) GTSRB

(b) CIFAR-10

Figure 7: **All-to-all attack**: Backdoor inputs and their corresponding heatmaps.