[Reviews · NeurIPS 2020]

Review 1

Summary and Contributions: The paper proposes an input-dependent trigger generation method for backdoor attacks. Cross-trigger loss is explicitly minimized so that one trigger is only effective in one image. This new method allows the attack to bypass a wide range of defense algorithms.

Strengths: The key innovation of the attack algorithm is well presented. The experiments well support the effectiveness of the attack.

Weaknesses: There lacks of comparison with existing dynamic backdoor attacks. The paper does not present how the proposed attack outperforms existing methods.

Correctness: Yes, the methods are correct.

Clarity: Yes, the paper is well written.

Relation to Prior Work: No clear discussion about previous backdoor attacks.

Reproducibility: Yes

Additional Feedback: The paper is clearly written and well demonstrates the effectiveness of the proposed attack. However, because of the ability of arbitrarily modifying the model, backdoor attacks are known to be hard to defend. Most backdoor attacks do not have an effective defense. In particular, the idea of using dynamically generated backdoor triggers to bypass defenses is not entirely new. For example [1] generates trigger patterns using auxiliary images, which results in creating non-repeating attacks, similar to this paper. Without comparing with existing backdoor attack methods and providing certain scenarios where the proposed method outperforms existing methods, it is difficult to conclude the contribution of this work. [1] Ji, Y., Liu, Z., Hu, X., Wang, P. and Zhang, Y., 2019. Programmable Neural Network Trojan for Pre-Trained Feature Extractor. arXiv preprint arXiv:1901.07766.


Review 2

Summary and Contributions: This paper proposes an approach to train a backdoor attack generator that can generate different backdoor triggers for different input, and the same trigger only works for a specific input. The author design unique evaluation metrics for the work. Experiments show that the proposed framework works on different datasets with different models.

Strengths: 1. The problem setting is novel. 2. The paper is well-written and the logic is clear. 3. Experiment result shows the backdoor attack has a very good successful rate.

Weaknesses: My major concern about this paper is the necessity of the proposed approach. The whole paper is based on the importance of dynamic triggers. However, as the authors mentioned, "dynamic triggers are hard to differentiate from adversarial noise perturbations". From this angle, it seems that any algorithm that efficiently generates adversarial sample can be served as a dynamic backdoor algorithm, with no poisoning samples inserted, and there has been a very rich literature about how to generate adversarial samples. The difference is that this paper requires the backdoor of each input to be different. However, because the input space is large, adversarial samples trivially satisfy the requirement. If the authors believe their dynamic backdoor has unique value, they should provide more justification against this trivial substitution. Another concern is that the added backdoor is very visible according to Figure 4. -------------------------------------- After rebuttal The authors briefly answers my concerns. However, I'm still unconvinced about the realistic value of the method, which is the reason that I ask those questions. The threat model of this paper is stronger than the threat model of evasive attacks such as adversarial samples, so I will hope to see the method can achieve stronger result. However, while the attack success rate is good, the sample generated is very visible, and also the process is costly compare to some lightweight attack models.

Correctness: Yes, I think the method is sound.

Clarity: Yes.

Relation to Prior Work: Yes.

Reproducibility: Yes

Additional Feedback: I believe this paper has a wrong subject area. It should be in "Social Aspects of Machine Learning -> Privacy, Anonymity, and Security".


Review 3

Summary and Contributions: This paper proposed a novel backdoor attack, which uses an input-aware dynamic trigger pattern through an input-conditioned generator during model training. The resulting backdoor models are more stealthy and difficult to mitigate the adversarial effects. The proposed attack can better evade several model and test-time defenses as well as vision interpretability tools, including Fine-Tuning, Neural-Cleanse, STRIP, and GradCAM. The key invention includes the novel use of diversity and uniqueness in generating trigger patterns Overall, the proposed attack is novel and is a stronger backdoor attack. The findings that its dynamic trigger pattern weakens many defenses and detection methods suggest a new and stronger attack for robustness evaluation, a big leap beyond using a universal trigger pattern.

Strengths: 1. Novelty: the proposed backdoor attack uses input-specific dynamic trigger pattern instead of conventionally used universal trigger pattern. The proposal of diversity and uniqueness (nonreusability) is also pretty novel. 2. The results of weakening several defenses give significant contributions. This attack has the potential of serving as a new baseline for robustness evaluation against backdoor attacks. 3. The experiments are thorough and convincing.

Weaknesses: I hope to see the following updates and discussions in the rebuttal, and I am happy to increase my review rating once my concerns are addressed. 1. In the ICLR publication "Bridging Mode Connectivity in Loss Landscapes and Adversarial Robustness", it was shown that Fine-Tuning is not the most effective approach to recover backdoor models, unless one has a sufficient amount of clean data to alleviate the backdoor effect. I would like to see how well the proposed attack is against the mode connectivity based defense proposed in the ICLR paper. 2. When reporting the attack/clean accuracy, it was unclear whether the authors were using data samples (and how many) from training data or testing data (unseen when training the backdoor model). I also would like the authors to address the robustness of the proposed approach when performing backdoor attack on testing (unseen) data. Intuitively, universal trigger pattern should be more robust to distribution shifts between training and testing data, and the input-aware apporach can be more sensitive (and cause attack to fail) if the generator cannot overcome the inherent distribution shift. Also, using testing (untrained) data makes more sense in all experiments. I hope the authors can clarity the data setting. 3.The probabilistic backdooring during model training according to Eq. (2) is worthy of more exploration. The authors only used rho_a=rho_c=0.1 in the experiments. I hope to see some parameter sensitivity analysis on these two parameters.

Correctness: Mostly correct and clearly explained in the supplementary material (there are some referencing issues in the supplementary document). The authors need to veritfy the reported results are using training data or testing (unseen data during training), and are suggested to test against more advanced defenses.

Clarity: This paper is well written and easy to follow.

Relation to Prior Work: The authors are suggested to test the robustness against more recent defenses, such as "Bridging Mode Connectivity in Loss Landscapes and Adversarial Robustness".

Reproducibility: Yes

Additional Feedback: Post-rebuttal comments: I thank the authors for providing additional results and clarification. My questions have been fully addressed, and I will increase my overall score. I hope the authors can incorporate the new results in the revised version.

[Author Response · NeurIPS 2020]

We thank the reviewers for their efforts and suggestions.

**R1. There lacks of comparison with existing backdoor attacks.** In terms of methodology, we discussed the previous attack methods in Sec 2.2. We pointed out their common weakness of using universal backdoor trigger pattern(s). Our proposed method overcomes this problem by using a trigger generator targeting on diverse and non-reusable patterns.

Empirically, there is no direct way to compare our method to others, and no such comparison was presented in the previous attack papers [3,2,10]. However, we can compare them indirectly by examining with backdoor defenses. The popular attacks such as BadNets and TrojanAttack can be detected by NeuralCleanse, as reported in that paper, while ours can surpass it. As for recent methods, we re-implemented them on the GTSRB dataset and examined them with GradCAM. The results are shown in Fig. 1a with Dynamic Backdoor [10] (top row) and Programmable Attack (bottom row). Unlike our method, both get caught by GradCAM since the classification prediction depends entirely on the small backdoor region. We will add this result to our revised paper.

**R1. The idea of dynamic backdoor is similar to Programmable Attack.** In that work, the trigger pattern is generated from the target image; it is still independent of the source image content. A pattern generated on one target image can be reused on any input image. Hence, it is still vulnerable to many defense methods. For example, GradCAM can spot the trigger as shown in Fig. 1a. Instead, our paper proposes generating non-reusable trigger patterns conditioned on the source images, achieved by novel constraints, leading to the capability to dodge all the mentioned defense mechanisms.

**R2. Any algorithm that efficiently generates adversarial sample can be served as a dynamic backdoor algorithm.** There are several differences between adversarial attacks and the proposed backdoor method. First, in adversarial attacks, we have no control over the deep model, making attack success uncertain. That model may be trained to adversarially robust to a wide range of attack algorithms. In backdoor attacks, we have total control on the model, guaranteeing a near 100% attack success rate. Second, adversarial attacks rely on expensive optimization processes. The computation is even more costly in black-box configurations. With our method, the attack process is fast and straightforward by running GPU-accelerated generators. Finally, due to universal adversarial examples, the adversarial generation on different input images can sometimes converge to the same adversarial pattern, violating our requirement. We will clarify this in our revised paper.

**R2. The added backdoor is very visible according to Fig. 4.** While the backdoor patterns are apparent to humans, like most of the previous backdoor attacks, they are still effective in fooling automated systems. As discussed in Sec 5, we plan to make them more realistic and imperceptible to humans in future studies.

**R3. Can the proposed attack surpass the defense in the Bridging Mode Connectivity paper?** We ran that method on our backdoor models on CIFAR-10 and reported the error rates in Fig. 1b. As can be seen, backdoor error rate is close to the corresponding clean error rate. The largest gap between them is only 20% when using 2500 training samples and t = 0.5. Hence, this defense method cannot mitigate our backdoor.

**R3. Unclear if the reported results are from training or testing data.** All of our results were computed on testing data, and details of data splits were reported in the supplementary PDF. It means our backdoored classifier, just by studying on training data, can recognize unseen backdoor patterns generated from unseen test images. This generalization ability is, in fact, a surprising and very important feature of our method. We will highlight it in the final version if this paper gets accepted.

**R3. Ablation studies on $\rho_a$ and $\rho_c$.** We report the results on CIFAR-10 in Fig. 1c. Overall, our method is quite stable with high accuracy on clean, attack, and cross tests. When increasing $\rho_c$, the cross accuracy increases from 80% to 93%. When increasing $\rho_a$, the attack success rate goes up to near 100%, and the cross accuracy also surprisingly increases.

(a)　　　　　　　　　　　　　　　(b)　　　　　　　　　　　　　　　(c)

Figure 1: Extra experimental results

[Meta-Review · NeurIPS 2020]

The reviewers for this paper are either on the fence about its merits or in favour of acceptance. While normally, this would make for a borderline paper, there are a few factors that give me confidence it is safe to include in the conference. All reviewers agree the method is novel, and the results and interesting. The outstanding concerns regarding the comparison are, I believe, decently addressed in the rebuttal. Even if they are not, the novelty of this method means it will at least provide solid grounds for discussion for people interested in adversarial and backdoor attacks. On the basis of the reviews, discussion, and rebuttal, I am happy to take a punt on an imperfect but interesting paper and recommend it is accepted. Pros: * Interesting and novel method * Impressive results * Room for improvement, but proves the concept